# Energy-Aware Edge Infrastructure Traffic Management Using Programmable Data Planes in 5G and Beyond

**DOI:** 10.3390/s25082375

**Published:** 2025-04-09

**Authors:** Jorge Andrés Brito, José Ignacio Moreno, Luis M. Contreras

**Affiliations:** 1Departamento de Ingeniería de Sistemas Telemáticos, ETSI de Telecomunicación, Universidad Politécnica de Madrid, 28040 Madrid, Spain; joseignacio.moreno@upm.es; 2Telefónica Innovación Digital, 28010 Madrid, Spain; luismiguel.contrerasmurillo@telefonica.com

**Keywords:** programmable data planes, SDN, energy proportionality, energy efficiency, traffic management, 5G and beyond, edge computing, P4, green computing

## Abstract

Next-generation networks, particularly 5G and beyond, face rising energy demands that pose both economic and environmental challenges. In this work, we present a traffic management scheme leveraging programmable data planes and an SDN controller to achieve energy proportionality, matching network resource usage to fluctuating traffic loads. This approach integrates flow monitoring on programmable switches with a dynamic power manager in the controller, which selectively powers off inactive switches. We evaluate this scheme in an emulated edge environment across multiple urban traffic profiles. Our results show that disabling switches not handling traffic can significantly reduce energy consumption, even under relatively subtle load variations, while maintaining normal network operations and minimizing overhead on the control plane. We further include a projected savings analysis illustrating the potential benefits if the solution is deployed on hardware devices such as Tofino-based switches. Overall, these findings highlight how data plane-centric, energy-aware traffic management can make 5G-and-beyond edge infrastructures both sustainable and adaptable for future networking needs.

## 1. Introduction

In recent years, programmable data planes have emerged as a prominent technological trend in the field of software-defined networking (SDN). Unlike traditional networking equipment, where control and data planes are typically coupled, programmable data plane devices provide flexibility by decoupling these two elements. This allows for the direct manipulation of packet handling at the hardware level, which is especially useful for adapting network behavior to meet diverse requirements. In this way, networks can be fine-tuned to meet different needs without requiring expensive equipment upgrades, a major shift in how networks are managed.

The specific applications for programmable data planes are wide-ranging and continue to grow [1,2,3,4]. Some notable use cases include in-band network telemetry (INT), network function virtualization (NFV), time-sensitive networking (TSN), firewalls, and traffic management.

In the context of 5G-and-beyond networks, programmable data planes are proving to be a key enabling technology for implementing and enhancing these architectures [5]. Fifth-generation-and-beyond services should accommodate a wide variety of usage scenarios, including enhanced mobile broadband, massive machine-type communication, and ultra-reliable low-latency communication [6]. In this regard, data plane programmability facilitates network function offloading, which helps reduce latency and enhances system performance by handling traffic in specialized hardware, such as application-specific integrated circuits (ASICs) or smart network interface cards (SmartNICs), which are optimized for high-speed packet processing. On the other hand, these programmable devices also allow operators to implement customized traffic management schemes, making it possible to optimize resources in dynamic network conditions while maintaining the quality of service (QoS) required by different applications.

5G and beyond is also driving significant growth in edge infrastructure. Edge computing aims to bring computational capabilities closer to end users and devices, thereby significantly reducing latency [7]. This proximity is critical for many delay-sensitive services, including augmented reality, autonomous driving, and industrial automation. Furthermore, because edge infrastructure provides computing and storage resources close to the user, tasks can be offloaded from centralized data centers, reducing the burden on core networks. With this distributed model, the overall efficiency and responsiveness of 5G-and-beyond networks can be improved.

Despite the numerous benefits and capabilities of next-generation networks, one of the main challenges they face is high energy consumption. The infrastructure required to implement 5G-and-beyond services includes a large number of base stations, edge servers, and network devices, which together consume a significant amount of energy [8]. This energy demand presents a significant concern with implications for both economic viability and environmental impact. Addressing this issue is necessary for making 5G and beyond sustainable, and for minimizing the carbon footprint associated with the deployment of this kind of technology [9,10]. The increased energy consumption also poses a challenge for telecom operators, as it directly affects operational costs [11,12,13]. Hence, there is a growing emphasis on developing energy-efficient solutions that can help optimize the power usage of critical infrastructure without compromising performance [14,15].

In response to these challenges, this paper proposes a data plane-centric traffic management approach coupled with an SDN controller that includes a dynamic power management module. Unlike previous energy-aware schemes that primarily rely on control plane decisions or use static device shutdown, our solution leverages programmable switches to conduct in-band traffic monitoring and load balancing, allowing the data plane to autonomously adapt to changing conditions. The controller, in turn, interprets the data plane insights to selectively activate or deactivate switches, achieving energy proportionality by aligning resource usage with real-time traffic demands. This combined strategy significantly reduces the overhead often associated with frequent control plane operations. Additionally, through experiments on multiple urban traffic clusters, we demonstrate how dynamically shutting down underused switches can effectively optimize network device usage. To reinforce these findings, a projected savings analysis estimates the potential reductions in operational costs and energy usage for a 5G-and-beyond edge infrastructure, underscoring the viability of our proposed scheme as a key contributor to future green networking initiatives.

The remainder of this article is structured as follows: Section 2 offers a theoretical overview of the key technologies, including SDN, programmable data planes, 5G and beyond, and edge computing. Section 3 reviews and analyzes related work. Section 4 describes the motivation, architecture, and methodology of our proposal, followed by Section 5, which details the experimental setup. Section 6 evaluates the resulting findings, and Section 7 provides a projected savings analysis for our scheme. Finally, Section 8 concludes this paper and outlines future work.

## 2. Technological Background

This section presents an overview of the technological foundations relevant to the proposed scheme, focusing on both the enabling technologies and the specific use case scenario. This approach is built on leveraging SDN with programmable data planes, while the practical application targets a 5G-and-beyond edge infrastructure.

### 2.1. Software-Defined Networking

Software-defined networking (SDN) is an emerging networking paradigm that decouples the control plane from the data plane, enabling centralized management and the programmability of networks. Unlike traditional networking models where each device handles control and forwarding independently, SDN separates these functions, allowing for more flexible and agile management. This separation is achieved by utilizing a centralized controller that communicates with network devices such as switches and routers through standardized protocols. The core principles of SDN [16] include the following:Separation of control and data planes: The control plane, responsible for deciding how packets are forwarded, is separated from the data plane, which carries out the actual packet forwarding. This decoupling provides greater flexibility and scalability for managing the network.Centralized control: A centralized SDN controller oversees the network’s control logic, enabling better decision-making. This controller can run on commodity hardware, allowing for cost-effective network management.Programmability: SDN allows the network to be programmed through application programming interfaces (APIs) that run on top of the SDN controller. This makes it possible to automate and dynamically configure network functions.Network virtualization: SDN facilitates network virtualization, allowing multiple virtual networks to share the same physical infrastructure.Use of open standards: SDN relies on open standards and protocols, such as OpenFlow [17], to ensure interoperability between different vendors’ products, enabling a more open and innovative ecosystem.

An SDN architecture can be represented as consisting of multiple layers, each with specific roles, as illustrated in Figure 1. The management plane (or application layer) includes network applications and services, such as MAC learning, intrusion detection systems, and routing algorithms, which communicate their needs to the control layer through the open northbound API. The control plane (or control layer) contains the SDN controller, which functions as the network’s “brain” processing these needs and deciding how data packets should be handled. Using the open southbound API, the control plane then sends instructions to the data layer. The data plane (or data layer) is composed of physical network devices, like switches and routers, which execute the instructions by forwarding packets accordingly.

### 2.2. Programmable Data Planes and P4

In a typical network setup, the data plane is responsible for processing and forwarding packets from one port to another. Traditionally, this functionality is fixed, meaning that any changes to how data are handled require new hardware or proprietary firmware updates. Programmable data planes change this by enabling users to write programs that define how packets should be processed, classified, or forwarded. These programs are written in domain-specific languages (e.g., P4 [18], Domino [19], and Lucid [20]), which provide operators with a high level of control over packet manipulation. Programmable data planes complement SDN because they decouple the network’s control logic from its data forwarding operations. By making the data plane programmable, it is possible to implement network functions directly within the hardware, such as custom packet forwarding, load balancing, INT, and advanced traffic engineering. This level of flexibility allows networks to quickly adapt to changing requirements without replacing hardware, thereby reducing costs and optimizing resources.

The Protocol-Independent Switch Architecture (PISA) is a pipeline-based model for data plane programmable devices that allows for detailed control over packet processing. This architecture [21] includes three programmable modules: a parser, a match–action pipeline, and a deparser, all working in coordination to manage incoming packets, as depicted in Figure 2. The parser handles the extraction of headers from arriving packets and parses them according to standard or custom protocols. It functions like a state machine providing detailed control over packet processing. At the core of the architecture is a match–action pipeline, which operates on packet headers using a series of match–action tables (MATs). This pipeline is organized into multiple stages that sequentially process and forward headers, allowing simultaneous operations across multiple memory blocks and Arithmetic Logic Units (ALUs). The deparser then reconstructs the packet, combining the processed headers and preparing them for transmission, ensuring the packet is properly serialized and ready for forwarding.

P4 (Programming Protocol-Independent Packet Processors) is the most widely adopted language for defining the behavior of the data plane [2]. Designed as a high-level, domain-specific language, P4 adopts the PISA processing pipeline to program packet processing. The language is both protocol-independent and target-independent, meaning that packet processing can be defined without being restricted to specific network protocols, and the code can be compiled to run on different network devices such as switches, routers, and programmable NICs, regardless of the hardware platform [18]. The development and standardization of P4 have been overseen by the P4 Language Consortium [22], now part of the Linux Foundation, and has gone through several iterations. The first standardized version, P4_14_ [23], was followed by P4_16_ [24] in 2016, which is the most recent version and significantly expanded the language’s capabilities, extending its applicability to a wider range of targets, including ASICs and SmartNICs.

Finally, P4Runtime [25] is a southbound API intended for enabling control plane communication with programmable data planes defined by a P4 program. It allows SDN controllers to configure and manage packet processing on network devices, independent of the specific hardware or protocols used. P4Runtime offers features such as updating tables, programming flows, and managing device states.

### 2.3. Fifth-Generation-and-Beyond Edge Technology

3GPP has been developing the 5G technology through Releases 15, 16, 17, and 18 [26]. Compared to previous versions, 5G aims for improved data rates, broader coverage, greater reliability, lower latency, and enhanced scalability. These improved features support a variety of new applications, referred to as verticals, such as immersive gaming, eHealth, autonomous vehicles (V2X), smart grids, and Industry 4.0. The IMT-2020 recommendation by ITU-T [6] categorizes these heterogeneous vertical environments into three main usage scenarios: Enhanced Mobile Broadband (eMBB), which delivers high data rates for multimedia access; ultra-reliable and low-latency communications (uRLLCs), which ensure dependable connections with high availability; and Massive Machine-Type Communications (mMTCs), which support connectivity for numerous devices with low-volume data transmission. The usage scenarios are linked to several Key Performance Indicators (KPIs) [27]. These KPIs specify the minimum technical requirements that must be achieved for a network to qualify as 5G-compliant. An overview of these requisites is provided in Table 1:

To fulfill these requirements, 5G incorporates novel paradigms like SDN, network function virtualization (NFV) [28], network slicing (NS) [29], and edge computing.

Edge computing [30] is a distributed computing approach that brings cloud capabilities closer to end-user devices by positioning computing resources at the edge of the network. By deploying services and caching content locally, edge computing helps to alleviate congestion in core networks, enabling more efficient local service delivery. These edge resources can be strategically deployed at locations such as base stations, network aggregation points, and edge data centers. Moreover, edge computing significantly reduces network latency, optimizes bandwidth usage, and supports applications requiring real-time data processing and ultra-low latency. As a result, it can play a significant role in fulfilling the stringent KPIs of 5G. Within a 5G architecture, edge computing can be fully integrated as a service that interacts seamlessly with other network functions [31].

In a service-based architecture (SBA) [32], the 5G system (5GS) provides essential functionalities to mobile networks by interconnecting a set of services. These services, known as network functions (NFs), fulfill distinct roles within the architecture. Some NFs of the 5GS include the following:Access Gateway Function (AGF): allows users from a wireline access network (wAN) to receive services provided by the same core network that supports mobile subscribers.User Plane Function (UPF): manages user data traffic by routing and forwarding packets. It also handles encapsulation and decapsulation, QoS management, and session statistics.Access and Mobility Management Function (AMF): handles signaling for authentication, connection and mobility.Session Management Function (SMF): oversees session management tasks, including the creation, modification, and termination of user data sessions.Authentication Server Function (AUSF): controls the authentication of a 3GPP or non-3GPP access.Network Repository Function (NRF): serves as a central registry of all NFs, allowing NFs to register and be discovered by other NFs.Policy Control Function (PCF): sets unified policy rules for control network functions, such as mobility, roaming, and network slicing.Unified Data Management (UDM): stores subscriber data and user profiles for use by the core network.

Sixth-generation technology, currently in the research phase, aims to build upon and exceed 5G capabilities [33]. These advancements will support groundbreaking applications such as immersive extended reality (XR), high-fidelity holographic communication, and digital twins, which are beyond the reach of current networks [34]. These new applications will introduce new KPIs, emphasizing precision, adaptability, and efficient automation [35]. Key enabling technologies for 6G include artificial intelligence (AI), terahertz communication, spectrum sharing, and innovative network architectures like converged RAN–Core and subnetworks. These technologies will significantly enhance communication networks, facilitating novel use cases that span diverse industries and pushing the boundaries of next-generation connectivity. Furthermore, 6G networks are expected to enhance sensing accuracy and feature adaptable, intuitive end devices that require minimal power, establishing a new standard in network automation and efficiency [36].

## 3. Related Work

Recent studies have introduced novel techniques to enhance load balancing, congestion control, and traffic splitting within programmable data planes. The authors in [37] presented a load-balancing approach leveraging an innovative in-network recirculation mechanism designed specifically for lossless data center environments. This method actively mitigates packet reordering by dynamically redirecting traffic flows based on real-time network congestion metrics, ensuring efficient traffic distribution while preserving packet sequence integrity. Complementarily, the survey presented in [38] extensively analyzed Remote Direct Memory Access (RDMA) transport protocols within data centers, discussing their inherent benefits for low-latency communication and highlighting critical implementation challenges related to congestion management. Additionally, [39] introduced an adaptive weighted traffic splitting algorithm, implemented directly in programmable data plane switches. This algorithm rapidly adapts to changing network conditions by intelligently distributing traffic loads across multiple paths at line rate, thus achieving balanced resource utilization. Furthermore, the approach described in [40] presented a highly scalable data plane-centric load-balancing solution that dynamically selects optimal forwarding paths based on continuous monitoring of network utilization and traffic load indicators. Similarly, the authors in [41] proposed a programmable data plane approach that significantly accelerates the adaptive traffic splitting process. This scheme assigns unique hash boundaries to paths, enabling rapid updates and efficient, accurate traffic distribution adjustments directly within the data plane, considerably reducing flow completion time compared to conventional methods. Despite these advancements, these studies primarily concentrate on performance aspects such as latency reduction, congestion avoidance, and efficient load distribution. However, they do not explicitly address the potential for energy savings or achieving energy proportionality through the dynamic and adaptive management of network resources.

The concept of in-network computing involves offloading computational tasks, such as NFs, to network components like switches, routers, and SmartNICs, as defined in [42]. Programmable data planes can play a key role as enablers for in-network computing, as stated in [3]. In this context, the work in [43] proposes an architecture for 6G networks in which NFs are allocated to hosts equipped with high-performance switching ASICs for packet transmission. A virtualization environment supports application deployment and migration among network hosts, thereby enabling faster processing and power consumption optimization. However, this architecture has a high level of abstraction, and the article lacks a detailed assessment of how packet switching functions contribute to overall energy reduction.

Significant research has been conducted to enhance energy efficiency in next-generation networks using SDN architectures. The study presented in [44] proposes an SDN-/NFV-based approach to optimize energy consumption in a 5G integrated fronthaul and backhaul setup. By leveraging an SDN controller and energy management APIs, this architecture collects traffic and energy usage data from software-based switches to make informed decisions about power state management and traffic rerouting. On the other hand, ref. [45] introduces an energy-aware scheme that minimizes active links in 5G backbone networks using multiple SDN controllers. By collecting network topology information, these controllers perform traffic engineering on control and data plane flows through heuristic static and dynamic algorithms. The controller described in [46] gathers network data and reconfigures the forwarding tables to concentrate traffic on the smallest number of networking devices, thereby enhancing energy efficiency. Additionally, the study in [47] presents an energy-saving metric considering coverage areas and traffic flows for radio access nodes with varying sleep modes, within heterogeneous 5G networks with separated control and data planes. The authors in [48] proposed an architecture integrating public and private blockchains for peer-to-peer (P2P) communication between SDN controllers and IoT devices, incorporating an energy-efficient routing mechanism for file transfers. In contrast to these efforts, our work focuses on implementing energy-aware analysis and load balancing directly in the data plane, which reduces the load on the control plane and prevents bottlenecks regardless of traffic type. Moreover, our scheme takes advantage of programmable data plane devices, which can be faster and more energy-efficient compared to traditional CPUs or fixed-function network devices, as shown in [49,50,51]. Additionally, research presented in [52,53] demonstrates the significant cost and energy advantages of deploying SmartNICs for service and network function offloading in comparison to conventional x86-based server solutions.

More recently, the authors in [54] presented a link deactivation strategy aimed at reducing energy consumption during periods of low network usage, particularly targeting the backbone infrastructures of ISPs. Their approach relies on monitoring traffic patterns to temporarily disable selected links, optimizing power usage without negatively impacting network reliability or traffic delivery. In a similar vein, the study in [55] proposed an SDN-driven routing scheme designed specifically to lower power consumption in traditional IP-based networks. By analyzing network topology and device-level power characteristics, their approach identifies and safely deactivates redundant network elements while rerouting traffic efficiently. Complementing these approaches, the authors in [56] introduced a power-aware traffic engineering framework utilizing deep reinforcement learning to dynamically optimize router power states according to changing traffic conditions. By leveraging real-time telemetry data, their system intelligently manages network resources to maximize energy savings without compromising network performance or availability. While these studies collectively advance the state of the art in energy-efficient SDN solutions, there remains substantial opportunity for developing approaches that directly address the unique demands and traffic variations typical of 5G-and-beyond scenarios. In particular, methods capable of performing dynamic load balancing directly in the data plane without adding overhead to packet headers could more efficiently align energy consumption with rapid, fine-grained variations in 5G-and-beyond edge traffic. Our proposed solution addresses precisely this need, utilizing programmable switches to adaptively manage traffic flows at line rate, thus ensuring energy-efficient resource utilization without significant overhead.

When considering the role of programmable devices in energy-saving strategies, ref. [57] presents a load-balancing scheme using P4-based virtual switches for green data centers. These switches gather information about power availability from servers to make decisions regarding workload allocation. However, this scheme does not consider turning off unused network devices, which could result in substantial energy savings, as demonstrated in [58,59]. Additionally, studies like [60,61] propose dynamic load-balancing schemes aimed at minimizing power consumption within data centers and 5G edge infrastructures by consolidating switch traffic and server workloads. These architectures employ programmable devices to carry out traffic engineering directly in the data plane, with the control plane intervening only at the initial stages of the process. Building upon these efforts, our approach introduces an SDN controller equipped with a dedicated dynamic power management module capable of adaptively shutting down inactive switches, thus achieving substantial reductions in energy consumption.

## 4. Motivation, Architecture, and Methodology

### 4.1. Motivation

The main motivation for the proposed traffic management approach is to achieve energy proportionality within an edge network infrastructure by dynamically adapting to fluctuating traffic demands. Energy proportionality refers to the ability of a system to adjust its energy consumption in direct proportion to its workload. In our scenario, this implies that as user traffic varies, the infrastructure must be able to adjust its power usage accordingly, thereby reducing energy waste. This is particularly relevant because network elements such as switches and routers are often statically provisioned, consuming significant power even during periods of low traffic demand [62,63].

Furthermore, our use case is centered on developing a dynamic edge infrastructure for a 5G-and-beyond UHD video streaming service traffic. By performing load balancing in the data plane, we aim to respond to changing traffic conditions, minimizing unnecessary overhead and preventing control plane bottlenecks. This dynamic methodology ensures that switches are only active when needed and powered down during periods of low activity. This technique emphasizes powering off inactive switches, which can be a promising and impactful energy-saving strategy [58,59]. To achieve this, we propose an architecture that integrates a programmable data plane with an SDN controller equipped with a dynamic power manager. The programmable data plane detects traffic variations and adjusts traffic flow accordingly, while the SDN controller determines which switches should remain active or be powered down based on the detected traffic conditions. This synergy between the data and control planes allows our system to adapt dynamically to network demands, thus aligning energy usage with actual network activity.

Although previous work has explored either data plane load balancers or control plane power managers (as discussed in the previous section), to the best of our knowledge, no existing solution combines both in a unified framework that can dynamically activate or deactivate switches in pursuit of energy proportionality in edge environments. By bridging this gap, merging data plane load balancing with a proactive, SDN-based power manager, our research aims to make 5G-and-beyond networks both responsive and energy-efficient. Leveraging the advanced capabilities of programmable data planes, along with a proactive SDN controller, we create an infrastructure that quickly adapts to the dynamic demands of emerging services while minimizing power draw.

Ultimately, our work addresses the need for more sustainable 5G-and-beyond systems by reducing overall power consumption. We aim to deliver a responsive, adaptable, and energy-efficient architecture that meets the evolving needs of next-generation edge networks.

### 4.2. Architecture

The proposed architecture is intended to support UHD video streaming services within a 5G-and-beyond edge infrastructure. This framework establishes service provisioning to a specific NF, that is, the AGF. In an SBA, the AGF acts as an intermediary between the physical access media (e.g., PON, GE, DSL) in the wAN and the core network [64]. Thus, the service hosted in the edge network will benefit from an AGF deployed near end-user devices. Furthermore, this function supports control and data plane separation [65], streamlining operations by employing a centralized control plane for configuration. Figure 3 illustrates the integration of the AGF with the edge network in a simplified SBA.

The edge network remains the primary focus of this work and is based on our original proposal in [61]. To optimize this infrastructure, we have adopted a spine–leaf topology, a widely used configuration in modern data centers. This topology is particularly effective in separating the control and data planes while efficiently managing network traffic distribution [66]. It provides a robust foundation for implementing load-balancing strategies and integrating programmable data plane and SDN controller configurations. The architecture and topology of our design are shown in Figure 4.

The hierarchical topology consists of three key components:Leaf switches: these nodes connect edge servers to the network, acting as the access points for data traffic.Spine switches: responsible for aggregating traffic between the leaf switches and the AGF, these switches manage the flow of data across networks.AGF switch: positioned as the interface for user traffic to and from the AGF, this switch ensures seamless integration of the NF with the edge services.

A fundamental aspect of this architecture is its use of programmable switches that support the P4 programming language, along with P4Runtime for communication between the control and data planes. This programmability highlights the proposed design, enabling the direct implementation of load balancing within the data plane and the dynamic power management of switches. Additionally, the SDN controller integrates various key modules, which are also depicted in Figure 4. A detailed explanation of these modules will follow in the next subsection.

### 4.3. Methodology

The proposed methodology further extends the previous work in [60,61] by incorporating an SDN controller with a dynamic power management module for optimizing the energy consumption of inactive switches within the network topology. This section describes the implementation of the load-balancing mechanism in the programmable data plane and elaborates on the power management strategy integrated into the control plane. The key enabler technologies for this proposal are the P4 programming language, which allows data plane programmability, and P4Runtime, the southbound API that enables seamless communication between the control and data planes.

#### 4.3.1. Data Plane Load Balancing

The primary goal of the data plane load-balancing scheme is to consolidate traffic between edge servers and the AGF into the minimum number of spine switches. We achieve this using P4 registers to store traffic-related metrics and Equal-Cost Multi-Path (ECMP) hashing to distribute flows. Figure 5 presents a high-level view of the packet processing pipeline, which includes the following components:
Parser: inspects Ethernet, ARP, IP, and TCP/UDP headers, extracting relevant fields such as source/destination IP addresses and transport ports.Ingress Pipeline: employs P4 registers, some of which are fully managed by the data plane ((a), (b), and (c)), while others are initialized by the controller ((d), (e), and (f)). This arrangement enables fast local decisions without constant control plane intervention. In more detail, we use the following:
(a)Traffic Volume: tracks the cumulative amount of traffic (in bytes) observed at each switch.(b)Packet Timestamp: stores the initial time reference information for packets, facilitating traffic estimation within a specified time window.(c)Spine Switches: indicates the number of active spine switches needed to handle current traffic demands.(d)Traffic Thresholds: specifies traffic volume limits that prompt the pipeline to mark more or fewer spine switches as needed.(e)Measurement Window: defines the time interval duration for counting traffic before resetting the traffic volume estimation in (a).(f)Switch Type: identifies whether a device is a spine, leaf, or AGF.


Once these registers detect that traffic has surpassed or dropped below thresholds from (d), the Spine Switch Calculation action updates register (c) and adjusts the ECMP hash range. Concurrently, the pipeline consults the IPv4 table to determine the next-hop interface for each packet.

Deparser: headers are reassembled, and the packet is transmitted out of the appropriate port.

Figure 6 illustrates the three main stages guiding the P4-based load-balancing process more explicitly:

State loading: The data plane sets registers (a), (b), and (c), while the SDN controller initializes registers (d), (e), and (f). A classification process identifies flows directed toward spine switches.Traffic volume measurement: Each packet increases the traffic counting in (a) by its size, with the timestamp from (b) helping determine whether the measurement window in (e) has expired. Once this time interval ends, traffic stored in (a) resets, and the timestamp in (b) is updated.Dynamic load balancing: The Spine Switch Calculation action compares the observed traffic in (a) against the traffic thresholds from (d) to determine how many spine switches are required, and then writes that number into (c). It subsequently applies the ECMP action, which relies on a 5-tuple hash over the source and destination IP addresses, IP protocol, as well as the TCP/UDP source and destination ports. This hash is bounded by the current value of spine switches from (c), scaling how many distinct paths are available. Moreover, a CRC-based hash is computed, and its outcome selects which port to use. As traffic fluctuates, the pipeline updates (c) on the fly, ensuring flows are automatically rehashed among the valid number of spine switches.

#### 4.3.2. Control Plane Forwarding Table Writing

The SDN controller is responsible for installing IP forwarding rules into each switch data plane. After discovering the topology, the controller identifies the next hop for each destination (typically each host’s subnet) and populates the switch forwarding table accordingly. When a switch receives a packet, these rules determine the correct egress port and MAC address rewrite for delivery toward the final destination. In essence, this module ensures that every switch in the network knows how to reach all endpoints, enabling unicast packet delivery without manual configuration on each device.

#### 4.3.3. Control Plane Multicast Formation

In addition to unicast forwarding, the controller creates and configures multicast groups. These groups allow packets (e.g., for broadcast or specific replication scenarios) to be duplicated and dispatched to multiple ports simultaneously. Even in a virtual environment, configuring multicast at the data plane level can reduce overhead compared to naive broadcast. The SDN controller identifies which ports should receive replicated packets and then programs each switch, ensuring that essential traffic such as certain control protocols or multi-destination streams are efficiently delivered to all necessary endpoints.

#### 4.3.4. Control Plane Register Initialization

As part of the programmable data plane design, a number of P4 registers are exposed to the controller for runtime configuration. These registers store key parameters like traffic thresholds, measurement windows, and switch types (e.g., leaf, spine, and AGF). The SDN controller initializes and updates these registers upon startup to enable advanced functionality, such as load balancing, traffic measurement, or specialized actions based on packet sizes. By writing the initial values, the controller ensures that all switches begin operation under a coherent global policy, one that can be dynamically adjusted as network conditions change.

#### 4.3.5. Control Plane Dynamic Power Management

The dynamic power management module, implemented within the controller, is responsible for adaptively managing the activation and deactivation of switch interfaces to optimize energy usage based on traffic conditions. Since the experimental environment is virtual (more details in the next section) and software switches cannot be properly powered off, our scheme achieves equivalent behavior by activating/deactivating switch interfaces at the system level.

Algorithm 1 provides a simplified pseudocode of the module’s routine. In each cycle, the controller reads the *spine_switches* from non-spine devices (e.g., leaf or AGF switches). As mentioned in Section 4.3.1, register (c) stores how many active spine switches are needed to handle current traffic demands. If all spines are required, all are turned on; otherwise, only a subset in the index range [*spine_start* … *spine_end*] is activated, while the rest are turned off.
**Algorithm 1** Dynamic Power Management  **Inputs:**              *switches[]*: an array of all switches in the network              *spine_switch_type:* an integer index that differentiates the spine switch role from the leaf and AGF types              *switch_interface_states*: a dictionary mapping each switch with the current interface state (activated/deactivated)              *spine_start*, *spine_end*: integer indices defining the range of spine switches   **Procedure**: 1   **for** *sw* **in** *switches*:2             **if** readRegister(*sw*, “*switch_type*”) == *spine_switch_type***:**
3                  **continue**
4             *required_spine_switches* = readRegister(*sw*, “*spine_switches*”)5             **if** *required_spine_switches* == *spine_end*:6                   **for** *i* **in range**(*spine_start*, *spine_end*):7                         update_switch_interfaces(*switches*[*i*]*,*
**On***, switch_interface_states*)8             **else**
9                   **for** *i* **in range**(*spine_start*, *spine_end*): 10                       *neededCount* = *spine_start + required_spine_switches*11                       **if**
*i* >= *neededCount*:12                               update_switch_interfaces(*switches*[*i*]*,*
**Off***, switch_interface_states*)13                       **else**
14                               update_switch_interfaces(*switches*[*i*]*,*
**On***, switch_interface_states*)

In more detail, the controller processes every switch in the topology (line 1). When register (f) shows that the device is a spine switch (line 2), the controller skips it (line 3), since these switches do not manage their own power state. For non-spine devices (leaf or AGF switches), the controller retrieves *required_spine_switches* from register (c) (line 4), which indicates how many spine switches are necessary. If this value equals *spine_end*, the data plane is signaling that all spine switches are needed, and thus the controller activates every switch in [*spine_start* … *spine_end*] (lines 5–7). Otherwise, the controller calculates how many spines to keep active (*neededCount* = *spine_start* + *required_spine_switches*) and deactivates any with an index above that threshold (lines 8–14). To prevent redundant tasks, the module maintains a *switch_interface_states* record of each switch interface’s status (active or inactive). By checking these states, the controller avoids unnecessary actions such as turning off an already inactive interface or turning on one that is already active, thus ensuring efficient management.

## 5. Experimental Setup

To implement our proposed architecture and methodology, an emulated edge network was set up using Mininet 2.3.1b1 [67]. In this environment, we use virtual hosts, software switches, and an SDN controller to reproduce the topology from Figure 4. Each software switch runs the Behavioral Model version 2 (BMv2) [68], a switch prototype specifically designed to run P4 programs. Figure 7 presents a detailed depiction of our Mininet-based implementation.

As observed, this virtual network includes eight BMv2 switches and nine hosts: SW1–SW4 as leaf devices, SW5–SW7 as spine devices, and SW8 acting as part of our AGF node, along with H1–H8 as edge server hosts and H9 as an AGF server host. Each server host is assigned an IP and MAC address via a topology JSON file, which also contains the exact host–switch port mappings (e.g., H1–SW1, H3–SW2, etc.). Depending on the test scenario, any host can serve as a traffic generator (client) or a traffic sink (server). In addition, we deploy a Python-based controller (sdncontroller.py) that loads forwarding rules and monitors the dynamic power state of spine switches via P4Runtime.

Traffic generation is conducted via iperf3 [69], which is capable of producing both UDP and TCP traffic patterns. Because our goal is to emulate UHD video streaming service demand, we predominantly generate UDP flows for these experiments. In more detail, we use a dedicated traffic-generation process that serves as our automated flow generator. This process reads a sequence of bandwidth targets from an input file, each representing a distinct load condition to be tested. We produce several parallel UDP streams using iperf3, allocating segments of the overall throughput among multiple randomly chosen hosts in the topology. These streams operate concurrently, pushing traffic from one designated “client” host toward whichever “servers” have been selected for that load cycle. This approach ensures that, at any given moment, traffic can flow to multiple destinations, pushing the data plane to distribute load across the available links. By varying the bandwidth targets, the generator emulates changing demand scenarios, enabling the network to showcase its ability to balance flows and adjust the power states of devices accordingly.

## 6. Evaluation

Using our Mininet testbed environment, we carried out a series of tests to validate the proposed traffic management scheme. To mirror realistic 5G-and-beyond edge computing traffic, eight traffic profiles from the study in [70] were selected as inputs of the flow generator described in the previous section. These profiles represent user flows from distinct metropolitan areas and capture variability in usage across different urban contexts. Figure 8 illustrates the four chosen traffic clusters: residential, public transportation, business, and recreational. Each cluster incorporates data captured on Monday and Sunday, thus reflecting the typical shift in user activity between weekday and weekend periods.

Because of limitations in our simulation tools, particularly in handling large-scale traffic flows, we scaled the original traffic patterns from MB ranges down to KB. Likewise, to fit a 24-h day into a shorter experiment, each 10-s slot in real time was treated as one hour; the SDN controller thus polled traffic data every 10 s, which was set in the Measurement Window register. Traffic was sent from edge service servers to the AGF switch to reproduce the coverage of user demand of a particular service hosted in the servers; by default, only one spine switch was enabled initially. For simplicity, we used statically predefined values for the Traffic Thresholds registers in our Python-based controller, basing them on each urban cluster’s representative traffic volumes.

For the first experiment, we used scaled-down traffic patterns from a residential cluster on Monday and Sunday. The controller thresholds were set at 300 KB and 500 KB, respectively, to determine when to activate two or three spine switches. Figure 9 displays how the system adapts to different traffic curves for a Monday (Figure 9a) and a Sunday (Figure 9b). Although both patterns exhibit similar overall volumes, the scheme still captures subtle variations, particularly during the daytime increases from 08:00 to 20:00. 

At the start of the day (00:00), both patterns exceed the 300 KB threshold, thus enabling two spine switches. Flows subsequently drop in the early morning, reducing usage to one switch. Later, traffic volumes rise sufficiently to activate a second spine switch again and, by late afternoon, surpass 500 KB and trigger activation of the third switch. Notably, Sunday’s load grows faster during midday hours, prompting earlier use of the third spine switch. Overall, Sunday employs three switches for 35% of the day, whereas Monday only does so for 28%.

Next, we examined a public transportation scenario that typically shows lower flow levels, setting thresholds at 50 KB and 100 KB. Figure 9 compares switch activation on Monday (Figure 10a), where two peaks around 09:00 and 18:00 surpass 100 KB and require all three switches 40% of the time, to Sunday (Figure 10b), which lacks similar surges. On Sunday, all three switches run only during a brief three-hour interval; in 92% of the daily cycle, one or two switches sufficiently handle the remaining traffic. 

In the third setting, we focused on a business area. Thresholds were set at 150 KB and 350 KB. As shown in Figure 11, Monday’s load (Figure 11a) surpasses 350 KB around 09:00, triggering all three switches, and later falls in midafternoon, allowing the third switch to be deactivated. Conversely, Sunday (Figure 11b) never reaches 350 KB, so only two switches are needed at most. For the remainder of Sunday, the traffic remains low enough that one spine switch suffices; consequently, the second switch is active only 24% of the time.

Lastly, we used traffic from a recreational cluster, which exhibits volumes similar to the business area; therefore, thresholds were kept at 150 KB and 350 KB. From Figure 12, Monday and Sunday share generally comparable patterns, though Sunday’s load climbs somewhat higher between 11:00 and 17:00, and then declines earlier in the evening. In response, Monday (Figure 12a) uses three active switches 24% of the time, whereas Sunday (Figure 12b) does so for 28% of the day. Two switches remain active for 32% of Monday and 16% of Sunday.

The results indicate that our dynamic traffic management scheme adapts consistently to diverse real traffic patterns by activating and deactivating spine switches according to the flows observed, including those with relatively subtle changes. Table 2 and Table 3 show the percentage of time each scenario runs with one, two, or three spine switches (individually in the first table and in combinations in the second), thus highlighting potential energy proportionality. In the following section, we examine how these usage patterns, when deployed on actual hardware, can translate into measurable power consumption.

## 7. Projected Savings Analysis

In this section, we focus on the potential energy savings of our dynamic traffic management scheme when deployed with hardware switches. As a reference device, we use the Wedge100BF-32Q [71], which can function as either a leaf or spine switch and incorporates an Intel Tofino ASIC [72] supporting a PISA-based architecture for P4 programmability. For power consumption estimates, we rely on the measurements from [73], which indicate that a single Wedge100BF-32Q switch consumes about 145.55 W under typical operational factors (e.g., port utilization, traffic load, and P4 program functionality). While this is a simplified figure, it serves as a reasonable baseline.

We first consider the case in which all three spine switches remain active for an entire 24 h period. The energy consumption ($E_{C}$) in this scenario is given by the following:(1)$$E_{C}=E_{Wedge100BF-32Q}*t*\sum_{x=1}^{n}x$$
where $E_{Wedge100BF-32Q}=$ 145.55 Wh, t=24 (h), n=3 for the three spine switches in our topology. If all switches remain active for the full day, the resulting consumption is approximately 10.48 KWh.

Next, to capture the behavior of our dynamic approach, which selectively activates or deactivates switches, we define the proportional energy consumption ($E_{PC}$):(2)$$E_{PC}=E_{Wedge100BF-32Q}*t*\sum_{x=1}^{n}p_{x}$$
where $p_{x}$ is the fraction of the day each spine switch remains active, derived from the usage percentages listed in Table 2. The energy saved ($E_{S}$) by our scheme is then as follows:(3)$$E_{S}=E_{C}-E_{PC}$$

Applying these equations to each scenario provides the values summarized in Table 4, which details the daily energy savings by area and day. Table 5 extrapolates these values to monthly and yearly scales, indicating that the savings can reach the megawatt-hour range over the course of a year.

A factor to consider in any on/off scheme is network availability. Activation times, link failure detection, and device responsiveness all affect uptime. According to [74], Tofino-based switches achieve sub-millisecond (microsecond-scale) reaction times, enabling relatively fast switch activation. In addition, reliability parameters like Mean Time Between Failures (MTBF) and Mean Time to Repair (MTTR) play major roles in ensuring overall availability [75]. In prior work [61], we observed that a similar spine–leaf topology can maintain an annual downtime near one minute. Thus, while a tradeoff exists between energy gains and service continuity, this tradeoff can be managed with carefully chosen devices and a resilient topology.

## 8. Conclusions and Future Work

This paper introduced a traffic management scheme that leverages programmable data planes and an SDN controller with dynamic power management to achieve energy proportionality, ensuring network resources are allocated in line with traffic demands. Proof-of-concept experiments across multiple urban traffic clusters highlight the scheme’s key benefits: network device usage is optimized by deactivating spine switches that are not actively needed, and these measures yield significant energy savings, as demonstrated by both empirical findings and a hardware-based projection analysis. Notably, the data plane-centric design also minimizes overhead on the control plane by handling the majority of load-balancing and monitoring tasks within the switches themselves. Taken together, these findings demonstrate that consolidating traffic management within the data plane and adaptively controlling switch power states provides a feasible and scalable pathway toward more sustainable 5G-and-beyond edge infrastructures.

Future work will involve scaling the proposed solution to more complex network topologies, such as fat-tree architectures, and evaluating its performance under diverse traffic patterns, including variable workloads. Additionally, deploying and testing the scheme on physical hardware (e.g., Tofino-based switches and SmartNICs) will help assess the impacts of frequent power cycling on network reliability, throughput, and latency. Furthermore, including operational and monetary cost analyses will quantify how dynamic switch management can bring both energy and financial benefits over extended periods.

## Figures and Tables

**Figure 1 sensors-25-02375-f001:**
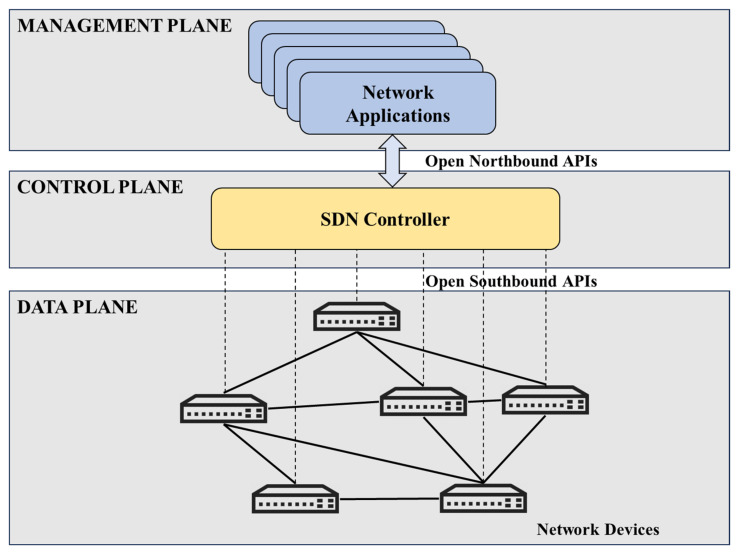
SDN architecture [16].

**Figure 2 sensors-25-02375-f002:**
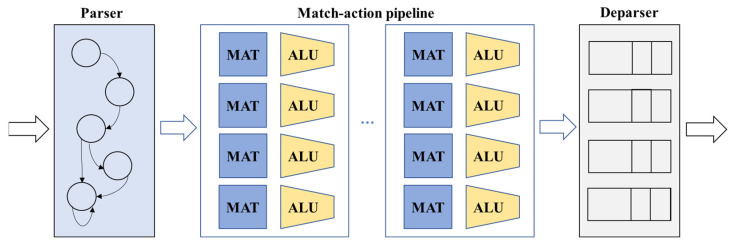
PISA programmable switch model [21].

**Figure 3 sensors-25-02375-f003:**
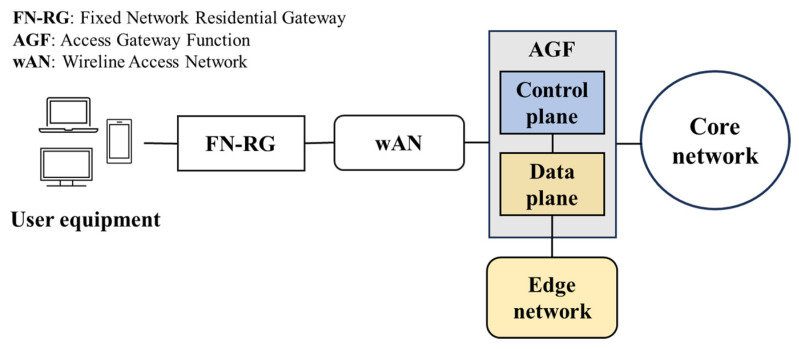
AGF and edge network integration in SBA.

**Figure 4 sensors-25-02375-f004:**
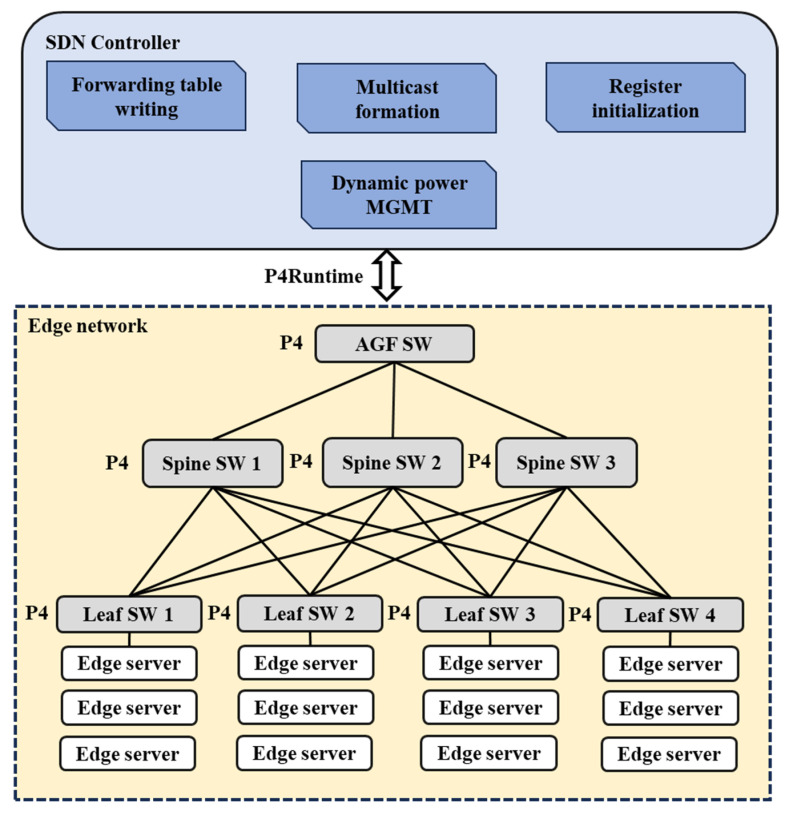
Proposed edge network architecture and topology.

**Figure 5 sensors-25-02375-f005:**
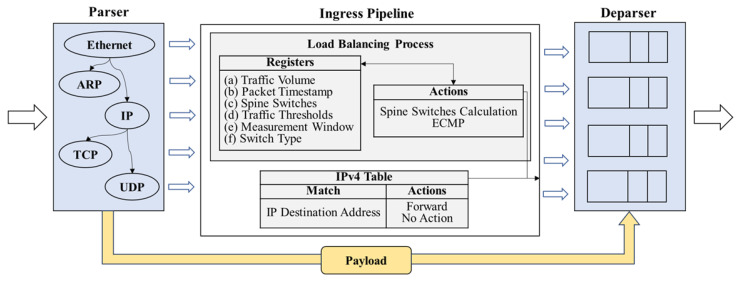
Data plane packet processing pipeline.

**Figure 6 sensors-25-02375-f006:**
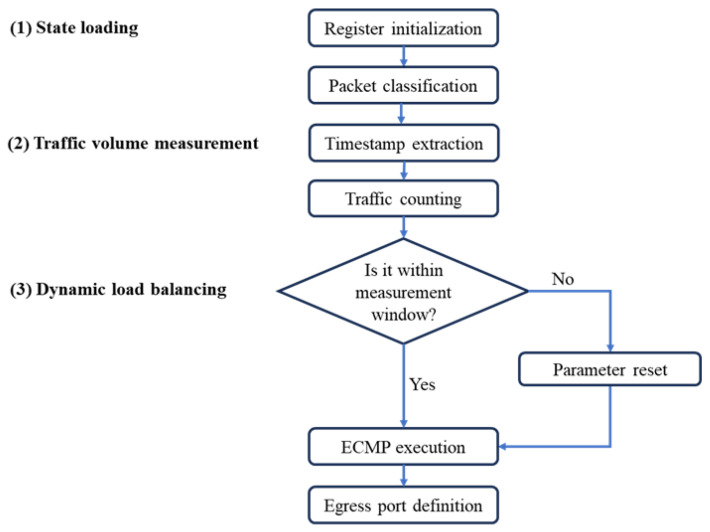
Data plane load-balancer flowchart.

**Figure 7 sensors-25-02375-f007:**
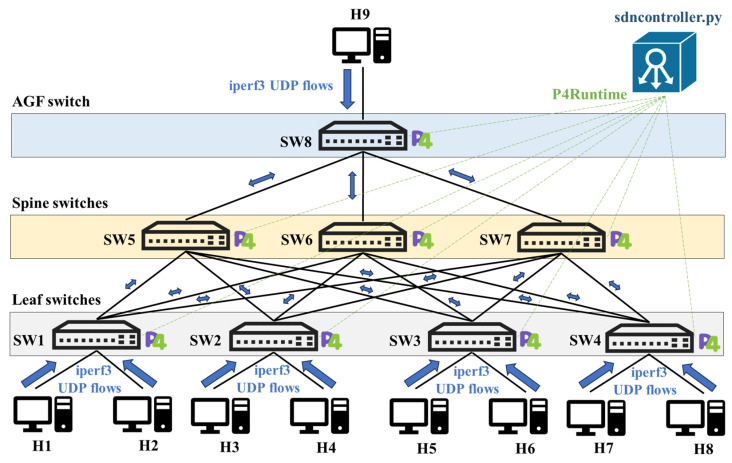
Mininet testbed environment.

**Figure 8 sensors-25-02375-f008:**
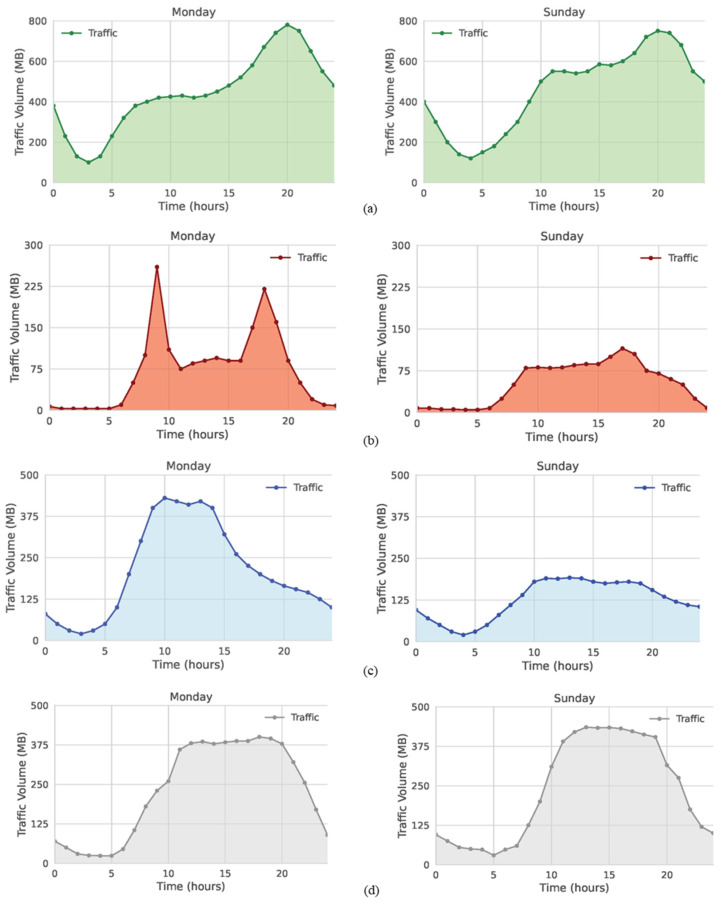
Traffic patterns of four urban clusters: (**a**) residential area; (**b**) public transportation area; (**c**) business area; (**d**) recreational area [70].

**Figure 9 sensors-25-02375-f009:**
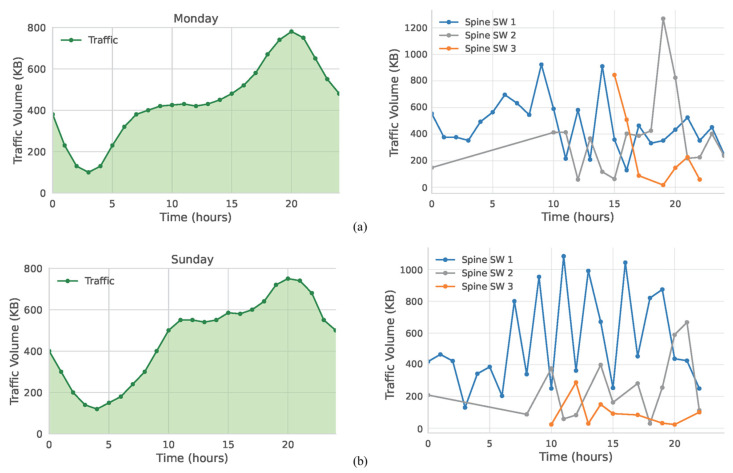
Traffic management of residential area for (**a**) Monday (weekday); (**b**) Sunday (weekend).

**Figure 10 sensors-25-02375-f010:**
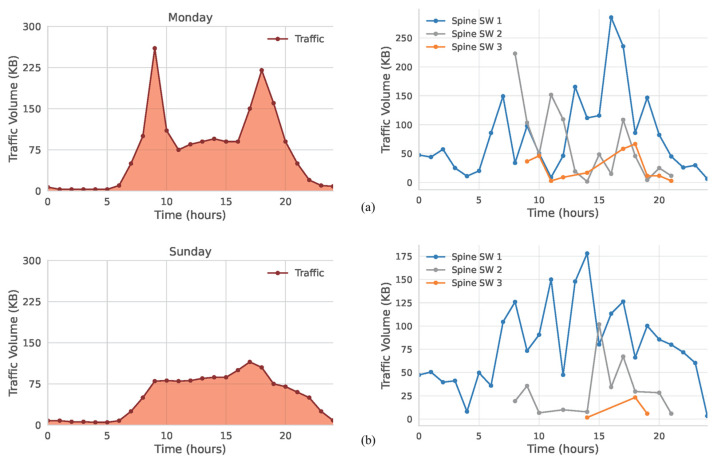
Traffic management of public transportation area for (**a**) Monday (weekday); (**b**) Sunday (weekend).

**Figure 11 sensors-25-02375-f011:**
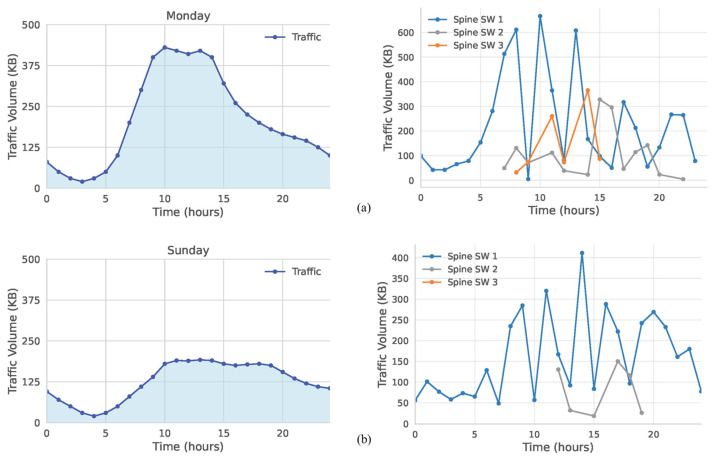
Traffic management of business area for (**a**) Monday (weekday); (**b**) Sunday (weekend).

**Figure 12 sensors-25-02375-f012:**
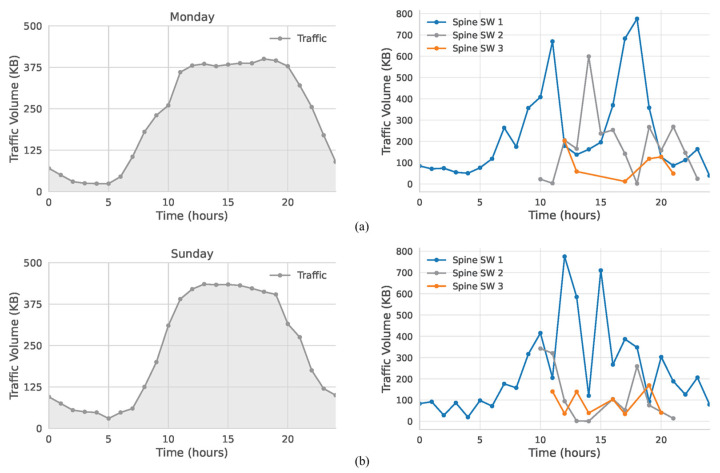
Traffic management of recreational area for (**a**) Monday (weekday); (**b**) Sunday (weekend).

**Table 1 sensors-25-02375-t001:** Minimum technical performance requirements for 5G [27].

KPI	Key Use Case	Values
Peak Data Rate	eMBB	DL: 20 Gbps, UL: 10 Gbps
Peak Spectral Efficiency	eMBB	DL: 30 bps/Hz, UL: 15 bps/Hz
User-Experienced Data Rate	eMBB	DL: 100 Mbps, UL: 50 Mbps (Dense Urban)
5% User Spectral Efficiency	eMBB	DL: 0.3 bps/Hz, UL: 0.21 bps/Hz (Indoor Hotspot); DL: 0.225 bps/Hz, UL: 0.15 bps/Hz (Dense Urban); DL: 0.12 bps/Hz, UL: 0.045 bps/Hz (Rural)
Average Spectral Efficiency	eMBB	DL: 9 bps/Hz/TRxP, UL: 6.75 bps/Hz/TRxP (Indoor Hotspot); DL: 7.8 bps/Hz/TRxP, UL: 5.4 bps/Hz/TRxP (Dense Urban); DL: 3.3 bps/Hz/TRxP, UL: 1.6 bps/Hz/TRxP (Rural)
Area Traffic Capacity	eMBB	DL: 10 Mbps/m^2^ (Indoor Hotspot)
User Plane Latency	eMBB, uRLLC	4 ms for eMBB and 1 ms for uRLLC
Control Plane Latency	eMBB, uRLLC	20 ms for eMBB and uRLLC
Connection Density	mMTC	1,000,000 devices/km^2^
Energy Efficiency	eMBB	Capability to support high sleep ratio and long sleep duration to allow low energy consumption when there are no data (e.g., above 6 GHz)
Reliability	uRLLC	1–10^−5^ success probability of transmitting a layer 2 protocol data unit of 32 bytes within 1 ms in channel quality of coverage edge
Mobility	eMBB	Up to 500 km/h
Mobility Interruption Time	eMBB, uRLLC	0 ms
Bandwidth	eMBB	At least 100 MHz; up to 1 Gbps for operation in higher-frequency bands

**Table 2 sensors-25-02375-t002:** Utilization percentage per spine switch by area and day.

Area	Monday			Sunday		
	Spine SW 1	Spine SW 2	Spine SW 3	Spine SW 1	Spine SW 2	Spine SW 3
Residential	100%	64%	28%	100%	57%	39%
Public transportation	100%	56%	40%	100%	44%	12%
Business	100%	54%	25%	100%	24%	0%
Recreational	100%	56%	24%	100%	40%	32%

**Table 3 sensors-25-02375-t003:** Utilization percentage per spine switch combination by area and day.

Area	Monday			Sunday		
	One Spine SW	Two Spine SWs	Three Spine SWs	One Spine SW	Two Spine SWs	Three Spine SWs
Residential	36%	36%	28%	39%	26%	35%
Public transportation	44%	16%	40%	52%	40%	8%
Business	46%	29%	25%	76%	24%	0%
Recreational	44%	32%	24%	56%	16%	28%

**Table 4 sensors-25-02375-t004:** Potential energy savings by area and day.

Area	Monday	Sunday
Residential	3.77 KWh	3.63 KWh
Public transportation	3.63 KWh	5.03 KWh
Business	4.23 KWh	6.15 KWh
Recreational	4.19 KWh	4.47 KWh

**Table 5 sensors-25-02375-t005:** Potential monthly and yearly energy savings.

Area	Monthly	Yearly
Residential	115.47 KWh	1.36 MWh
Public transportation	126.53 KWh	1.47 MWh
Business	150.33 KWh	1.74 MWh
Recreational	132.69 KWh	1.56 MWh

## Data Availability

The original contributions presented in this study are included in the article, while the raw data and supporting materials are available on request to the corresponding author.

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
