# Peer review of "Energy-Aware Edge Infrastructure Traffic Management Using Programmable Data Planes in 5G and Beyond"

_sensors, 2025, doi:10.3390/s25082375_

Round 1
Reviewer 1 Report
Comments and Suggestions for Authors
The paper presents a new traffic management mechanism in next-generation networks by integrating programmable data planes with SDN controllers. The proposed scheme integrates flow monitoring on programmable switches with a dynamic power manager in the SDN controller, which selectively powers off inactive switches. Thus, it achieves energy proportionality, especially in addressing the energy demands of 5G and beyond. However, it would be beneficial to clarify the design details and compare with the state-of-the-art energy management solutions.
1. The methodology employed for flow monitoring and the dynamic power management of switches is well-explained. However, it's necessary to have more details on the algorithm used for determining when to power off switches. Please provide more mathematical analysis for the thresholds and decision-making process.
2. Please explain specify how the switch processes packets, or the author can describe the pipeline for data packets in programmable switches by using figures for easy understanding.
3.The evaluation results show the significant energy savings, it's better to conduct more test to evaluate more metrics under realistic workloads.
4.It's necessary to add more state-of-the-art related works, e.g.,
J. Hu, Y. He, W. Luo, J. Huang , J. Wang. Enhancing Load Balancing with In-network Recirculation to Prevent Packet Reordering in Lossless Data Centers. IEEE/ACM Transactions on Networking, 2024, 32(5): 4114-4127.
J. Hu, H. Shen, X. Liu, J. Wang. RDMA Transports in Datacenter Networks: Survey. IEEE Network, 2024, DOI: 10.1109/MNET.2024.3397781
J. Hu, S. Rao, M. Zhu, J. Huang, J. Wang, J. Wang. SRCC: Sub-RTT Congestion Control for Lossless Datacenter Networks. IEEE Transactions on Industrial Informatics, 2024, DOI: 10.1109/TII.2024.3495759
Comments on the Quality of English LanguageThe paper is well-written, but certain sections could be improved for clarity. For example, the introduction could more clearly outline the specific research questions being addressed. Additionally, the conclusion should summarize the key findings more succinctly.
Reviewer 2 Report
Comments and Suggestions for Authors
This paper developed an energy-efficient traffic management framework for 5G and beyond networks, leveraging programmable data planes and a dynamic power management system controlled by SDN. The proposed system dynamically adapts to fluctuating traffic demands by selectively activating and deactivating network switches based on real-time traffic conditions. This energy proportionality approach ensures that energy consumption aligns closely with network usage, reducing unnecessary power expenditure while maintaining performance. Additionally, the manuscript provides a detailed architectural design and methodology for integrating this traffic management system into edge computing infrastructures, particularly for UHD video streaming services. The authors presented simulation results demonstrating the system's effectiveness in reducing power consumption and operational costs while maintaining network performance. Unfortunately, the authors should resolve the below concerns of the reviewer.
1) The manuscript built on existing work on energy-aware traffic management using programmable data planes in 5G and beyond but did not introduce fundamentally new techniques. Many of the concepts, such as SDN-based dynamic power management and programmable switches, have been explored in prior research, and the manuscript does not highlight clear advancements over existing approaches. Clarifying the novelty of the paper is critical for the acceptance decision.
2) The study relied heavily on simulations and an emulated environment (Mininet) rather than real-world hardware deployment. The absence of real hardware testing (e.g., on Tofino-based programmable switches) weakened the practical applicability of the proposed solution. Also, the proposed dynamic traffic management scheme was tested only on predefined urban traffic scenarios. However, a more diverse set of conditions (e.g., rural deployments or highly variable traffic environments) could provide better insights into its effectiveness.
3) The study focused on small-scale simulations with limited traffic loads. There was little discussion on how the system would perform under high traffic loads or in large-scale network environments. Furthermore, the paper did not adequately address the potential overhead introduced by frequent switch activation and deactivation on the network control plane.
Last but not least, the authors should clarify the novelty against conventional works for reconsideration of acceptance. A thorough literature survey is recommended.
Round 2
Reviewer 1 Report
Comments and Suggestions for Authors
Revised paper is fine and can be accepted now.
Author Response
Thank you for your review.
Reviewer 2 Report
Comments and Suggestions for Authors
The authors have resolved all comments of the reviewer so the paper is accepted for publication now.
Author Response
Thank you for your review.